# Efficiency of the Brain Network Is Associated with the Mental Workload with Developed Mental Schema

**DOI:** 10.3390/brainsci13030373

**Published:** 2023-02-21

**Authors:** Heng Gu, He Chen, Qunli Yao, Wenbo He, Shaodi Wang, Chao Yang, Jiaxi Li, Huapeng Liu, Xiaoli Li, Xiaochuan Zhao, Guanhao Liang

**Affiliations:** 1Center for Cognition and Neuroergonomics, State Key Laboratory of Cognitive Neuroscience and Learning, Beijing Normal University, Zhuhai 519087, China; 2School of Systems Science, Beijing Normal University, Beijing 100875, China; 3Institute of Computer Applied Technology of China North Industries Group Corporation Limited, Beijing 100821, China

**Keywords:** mental workload, mental schema, functional brain network, small-worldness, global efficiency, local efficiency

## Abstract

The study of mental workload has attracted much interest in neuroergonomics, a frontier field of research. However, there appears no consensus on how to measure mental workload effectively because the mental workload is not only regulated by task difficulty but also affected by individual skill level reflected as mental schema. In this study, we investigated the alterations in the functional brain network induced by a 10-day simulated piloting task with different difficulty levels. Topological features quantifying global and local information communication and network organization were analyzed. It was found that during different tests, the global efficiency did not change, but the gravity center of the local efficiency of the network moved from the frontal to the posterior area; the small-worldness of the functional brain network became stronger. These results demonstrate the reconfiguration of the brain network during the development of mental schema. Furthermore, for the first two tests, the global and local efficiency did not have a consistent change trend under different difficulty levels, but after forming the developed mental schema, both of them decreased with the increase in task difficulty, showing sensitivity to the increase in mental workload. Our results demonstrate brain network reconfiguration during the motor learning process and reveal the importance of the developed mental schema for the accurate assessment of mental workload. We concluded that the efficiency of the brain network was associated with mental workload with developed mental schema.

## 1. Introduction

The study of mental workload has attracted much interest in neuroergonomics, a frontier field of research. Mental workload is defined as “the portion of an individual’s limited mental capacity that is actually required by task demands” [1]. The increase in task demands necessitates additional cognitive resources. When the demands of the task are close to the mental capacity of the individual, known as mental overload, it may lead to deteriorated performance [2]. While numerous physiological measures have been proposed, there appears no consensus on their validity as effective agents of mental workload [3] because mental workload is not an intrinsic attribute of the human brain but the result of the interaction between the operation task and mental resources of the operator. It could be affected by factors such as task demands, skill level, and problem-solving strategies [4].

Training is a process of improving competencies or performance by acquiring or optimizing a series of inter-related movements for the goal. It can be observed that the individuals usually go through the following stages: first, they try to understand what is necessary to complete the task; second, they try to find a familiar solution or establish a new strategy to complete it; and third, they map the optimal solution to the task demands, followed by more accurate or efficient performance. Such an organized unit of knowledge to guide the current understanding or action is known as a mental schema [5]. It describes a mental framework that structures various categories of information and their relationships, as determined by past experience. Individuals are more likely to make mistakes if their mental schema is inadequate, especially when confronted with new situations [6]. Skill acquisition represents the development of a high to a low degree of dependence on conscious control of movements [7]. As such, the development of mental schema represents the increasing efficiency of cognitive processing required to plan and make movements.

In recent years, the study of neuroergonomics from the perspective of brain networks is attracting attention [8,9,10,11], and a growing number of studies have investigated the “efficiency” of brain networks in terms of the cost of transmitting information [12]. Small-world networks are of particular interest when studying human brain networks because of their high global and local information transmission efficiency, low energy and wiring costs, and their excellent suitability for complex brain dynamics [13]. The topology of a network is important to its overall function and performance [14]. Segregated processes (for example, visual processing) would benefit from highly clustered connections of a local region (e.g., the occipital cortex), whereas integrated processes (e.g., executive functions) would benefit from efficient information transfer throughout the network [14].

For the large-scale brain networks modeled from different types of human neuroimaging data, brain regions or sensors are usually used as nodes and anatomical or functional connectivity as edges. The electroencephalogram (EEG) is a non-invasive means of measuring electrical activity inside the brain and is capable of capturing the rapid temporal dynamics of brain activity and information flow among different regions at the sub-second time scale [15]. With its wearability and ease of use, EEG has become attractive and applicable to neuroergonomics studies [16]. Previous studies have elucidated the complex relations between cerebral regions involved in skill acquisition [8], motion [10,11], mental workload [17,18,19,20], and mental fatigue [21,22]. Recently, several studies have claimed that behavioral performance fluctuates depending on the phase synchronization between brain regions across different tasks and sensory modalities [23,24]. To this end, network metrics, such as global efficiency ($E_{G}$), local efficiency ($E_{L}$), and small-worldness (σ), have been proven to be significantly altered to meet task demands. Network efficiency measures the efficiency of information exchange in the network. Additionally, the small-world metric, defined by efficiency, precisely quantifies the degree to which a network has both higher local and global efficiency [25].

Studies on the application of network metrics in mental workload can be found in many previous works. Dimitriadis et al. [26] found that there was a decrease in theta local efficiency during difficult tasks in parieto-occipital regions. Huang et al. [18] also observed a decrease in theta local efficiency, but an increase in theta global efficiency. However, there were still other studies with inconsistent findings. Shaw et al. [27] found that efficiency in the theta band increased during a physical task across the brain, both locally and globally. There have also been a lot of studies to support the claim that the human brain exhibits a small-world network topology [19,28,29]. Furthermore, an interesting U-shaped pattern was also observed in the theta band by Porter et al. [30]. Heterogeneity in the findings could be attributed to a variety of factors, including experimental tasks, connectivity estimating techniques, threshold values, and so on. Overall, most studies using graph theory for analysis still concentrate on cognitive tasks under controlled conditions. Real-world studies on motor processing are limited, and long-term follow-up studies are rare. The purpose of this study is to explore how skill level development and task difficulty change the characteristic patterns of brain networks. To reach this goal, a quadrotor UAV (unmanned aerial vehicle) simulated flight training task is chosen. The weighted Phase Lag Index [31] is used to measure phase synchronization between sensors. Three network metrics (i.e., global efficiency, local efficiency, and small-worldness) are calculated from EEG signals collected during the task.

## 2. Materials and Methods

### 2.1. Participants

A total of 50 male volunteers (age = 23.25 ± 2.12 years) participated in this study. Participants were students recruited from the Beijing Normal University, none of whom had experience in operating a quadrotor UAV. This study was approved by the Ethics Committee of Beijing Normal University. All participants gave written informed consent before participation and underwent 10-day UAV flight training. A more detailed description is provided in a previously published article [32].

### 2.2. Experimental Protocol

The participants went through a series of training and testing sessions on the UAV operation described below, as shown in Figure 1. They were required to complete six task sessions (4 test sessions and 2 training sessions) over 10 days (days 1–4, day 7, and day 10), with one session a day. Each task session took less than 1 h/day. The UAV operation task is a computer-based flight simulation, in which they were asked to direct the UAV to fly over several waypoints arranged in an “figure eight (∞)” pattern. We defined four scenarios with different difficulty levels (L0: no wind; L1: fixed direction breeze; L2: fixed direction strong wind; L3: random direction strong wind) to induce different mental workload levels for the participants. During training sessions (day 2 and day 3), they were asked to operate for 1 h in the L0 scenario. During test sessions (day 1, day 4, day 7, and day 10), the participants were required to complete piloting tasks in ascending or descending order in 4 scenarios with a total of 16 trials (4 scenarios × 4 trials/scenario). For each flight, the subjects were required to complete the Cooper–Harper scale [33] to evaluate perceived workload. Before and after each task session, they were asked to perform the resting state task, consisting of two minutes each with their eyes closed and open.

### 2.3. EEG Acquisition and Preprocessing

EEG signals were recorded from 24 scalp electrodes at a sampling rate of 1 kHz while the participants were in the task session. The layout of the cap follows the international 10–20 system: frontal—Fp1, Fp2, F3, F4, and Fz; central—C3, C4, FC5, FC6, CP5, and CP6; parietal—P3, P4, and Pz; occipital—O1 and O2; left temporal: F7, T3, TP7, and P7; and right temporal: F8, T4, TP8, and P8. All the electrodes were grounded to Fpz and referenced to Cz, and the impedances were kept below 10 kΩ. For offline analysis, the signals were bandpass-filtered (0.5–48 Hz) and down-sampled to 250 Hz. The independent component analysis (ICA) [34] was used to remove artifacts associated with eye blinks and EMG by visual inspection. Then, the continuous EEG signals were segmented according to the event markers of different task scenarios. As the participant may request to restart the task due to the loss of control of the UAV, data segments less than 30 s were discarded. All preprocessing algorithms were performed using EEGLAB in MATLAB R2018b.

### 2.4. Functional Connectivity and Network Topology Analysis

To determine the individual alpha peak frequency (iAPF), the center of gravity (COG) [35] method was applied using the rest-state data of closed eyes before and after task sessions, and the mean values of the two data segments were used as the iAPF of the session.

Among all the EEG components, the theta band activities were mostly reported to be involved in the execution of tasks [36,37,38]. In order to track the changes in connectivity caused by the learning process and task difficulty, the functional connectivity was estimated in the individual theta frequency bands (4 Hz-0.8*iAPF) using the weighted Phase Lag Index (wPLI) between each pair of electrodes [31]:(1)$$wPLI=|\frac{\sum_{t=1}^{n}\left|imag(S_{xy,t})|\mathrm{sgn}(imag(S_{xy,t})\right)}{\sum_{t=1}^{n}\left|imag(S_{xy,t})\right|}|$$

In order to investigate changes in the topological properties term of information flow [39], global efficiency ($E_{G}$), local efficiency ($E_{L}$), and small-worldness (σ) were employed in this work using the Brain Connectivity Toolbox [40]. To compare the topological organization of the brain functional network among different conditions, we used the median of all scenario data for each test condition as the threshold to binarize the connection matrix.

For a graph *G* with *N* nodes, the characteristic path length, which is a measure of the overall communication efficiency between any pair of nodes, is computed as:(2)$$L=\frac{1}{N(N-1)}\sum_{i\in N}\sum_{i\neq j\in N}\min\{L_{ij}\}$$

The global efficiency ($E_{global}$) and local efficiency ($E_{local}$), which measured the efficiency of information exchange in the network, are computed as:(3)$$E_{global}=\frac{1}{N(N-1)}\sum_{i\neq j\in N}\frac{1}{\min\{L_{ij}\}}$$
(4)$$E_{local}=\frac{1}{N}\sum_{i\in N}E_{global}(G_{i})$$
where $E_{global}(G_{i})$ is the global efficiency of $G_{i}$, the subgraph consisting of the neighbors of node i, and $\min\{L_{ij}\}$ is the shortest path length between nodes i and j.

The clustering coefficient (*C*), which quantifies the degree of mutual connections between the nearest neighbors of a node, is computed as:(5)$$C=\frac{1}{N}\sum_{i\in N}\frac{2E_{i}}{\left(k_{i}(k_{i}-1\right)}$$
where $k_{i}$ is the number of edges directly connected with node i, and $E_{i}$ is the number of triangles around node i.

In order to describe the small-worldness properties, the normalized characteristic path length $\gamma =C/C_{random}$ and the normalized characteristic path length $\lambda =L/L_{random}$ are computed, where $C_{random}$ and $L_{random}$ represent the average clustering coefficient and average characteristic path length obtained from 100 surrogate random networks. These two metrics could be unified as a small-worldness metric, i.e., σ=γ/λ. A network is considered small-world if it meets the criteria: σ>1 [41].

### 2.5. Statistical Analysis

The behavioral data, subjective ratings, and network metrics were subjected to the 3 (scenarios: L0, L1, L2) × 4 (tests: T1, T2, T3, T4) repeated measures ANOVA with the test orders (Tests 1–4) and task difficulty levels (L0–L2) as independent variables. The data of the L3 scenario were not included in the analysis because the participants’ assessment of the difficulty level of this task was inconsistent (the details are presented in the Appendix A). Geisser–Greenhouse (G-G) corrections were used when sphericity violations occurred in the omnibus tests. Further post hoc analysis was conducted for the significant interactions using a paired t-test. The Bonferroni approach was used for multiple comparison corrections. The effect size was estimated using η_g_^2^ statistics. RStudio (version 1.4 for Windows) was used for statistical validation of the analysis results.

## 3. Results

### 3.1. Task Performance

Although after training, all subjects were able to complete the tasks in all scenarios in the last three tests (Test2–Test4), 26 subjects failed to complete the test task in the L2 scenario due to insufficient operational skills and scored 0 in Test1. Therefore, only the behavioral data of the remaining 24 subjects were analyzed to avoid the spurious enhancement of statistical significance by the L2 scenario scores of these 27 subjects. The average of all test scores in the same scenario was taken as the performance in that scenario.

The averaged performance scores across participants are shown in Figure 2A. The repeated measures ANOVA revealed the main effect of the difficulty level (F_2,46_ = 75.514, *p* < 0.001, η_g_^2^ = 0.332) and the test order (F_2.27,52.17_ = 44.322, *p* < 0.001, η_g_^2^ = 0.285). However, the interaction failed to pass the significance level criterion (F_6,138_ = 0.353, *p* = 0.907, η_g_^2^ = 0.005). The post hoc tests show that the performance was significantly different across conditions (L0 > L2, T1 < T4). These results suggest that the participants did feel the variation in task difficulty in different task scenarios, and as the training progressed, their operational skills improved.

### 3.2. Perceived Workload

The data of all subjects were included for the analysis. The average of all subjective scales in the same scenario was taken as the scale in that scenario. The averaged workload assessed by Cooper–Harper Scale across participants were depicted in Figure 2B. The repeated measures ANOVA revealed the main effect of the difficulty level (F_1.19, 60.62_ = 169.1, *p* < 0.001, η_g_^2^ = 0.152) and the test order (F_1.41,72.05_ = 51.39, *p* < 0.001, η_g_^2^ = 0.151). However, the interaction failed to pass the significance level criterion (F_3.25,165.99_ = 1.691, *p* < 0.167, η_g_^2^ = 0.001). The post hoc tests show that subjective rating was significantly different across conditions (L0 < L2, T1 > T4). These results suggest that the perceived workload was higher with increasing difficulty levels, and the repeated training reduced the perceived workload at the same difficulty levels.

### 3.3. Network Analysis Results

To grasp a brief overview of the dynamics of brain connectivity during the whole task session, the repeated measures ANOVA was performed for each channel pair in the connection matrix. The average connection strength of channel pairs with significant interaction effects (*p* < 0.05/276) is shown in Figure 3. As shown in the topographic map, channel pairs with significantly different connectivity strengths under different conditions were widely distributed across the brain. The connection strength was significantly higher on Test1 than on the other three tests (Test2–Test4), especially the long-range connections between the prefrontal, temporal, and posterior regions. Moreover, the overall connection strength increased with the difficulty levels in Test1, which was not observed in the other three tests (Test2–Test4).

To investigate the network efficiency of information integration and transmission, the repeated measures ANOVA was further employed for the global efficiency metric ($E_{global}$). The main effect of the difficulty levels (F_2.00,98.00_ = 9.739, *p* < 0.001, η_g_^2^ = 0.015) and the interaction between the test order and the task difficulty (F_4.22,207.02_ = 22.028, *p* < 0.001, η_g_^2^ = 0.101) were found to be statistically significant but not for the test order (F_2.37,116.04_ = 0.9125, *p* = 0.419, η_g_^2^ = 0.007). Post hoc tests were performed, as depicted in Figure 4A. These results demonstrate that during skill development (Test1–Test2), the global efficiency of the network did not have a consistent change trend under different difficulty levels, but when the subjects master the operation skills (Test3–Test4), the global efficiency will decrease with the increase in task difficulty, showing the importance of developed mental schema and the sensitivity of global efficiency to the increase in mental workload.

To further characterize the changes in the key regions of information transmission across the brain network, the detailed local efficiency ($E_{local}$) of each channel was analyzed. The repeated measures ANOVA was performed for each channel. There were significant interactions between the test order and the task difficulty in most of the channels (*p* < 0.05/24) except for CP5, P3, and TP7. For the main effect of test orders, only four channels (C4, F3, O1, and T7) failed the test of statistical significance; for the main effect of difficulty levels, only three channels (O1, O2, and T8) failed the test of statistical significance. The topographic maps of the local efficiency are shown in Figure 4B. In Test1, the frontal region showed higher local efficiency for all three different levels; however, there was no consistent change trend and distribution characteristics among different difficulty levels in Test2, whereas the network was in the process of dynamic adjustment. In contrast, for the latter two tests (Test3 and Test4), the key brain regions shifted from the frontal to the posterior regions, and the local efficiency decreased as the task difficulty increased, showing a stable spatial pattern and sensitivity to the increase in mental workload.

To investigate the reconfigurations of the brain network with different task difficulties across test orders, the repeated measures ANOVA was employed for the small-worldness metric (σ). The main effect of the test order (F_3.00,147.00_ = 3.556, *p* = 0.016, η_g_^2^ = 0.019) and the interaction between the test order and the difficulty levels (F_4.22,206.74_ = 2.693, *p* = 0.030, η_g_^2^ = 0.023) were found to be statistically significant but not for the main effect of task difficulty (F_2.00,98.00_ = 0.799, *p* = 0.453, η_g_^2^ = 0.003). Post hoc tests were performed, as depicted in Figure 4C. For scenarios with developed mental schema (L0 scenario), the small-worldness metric did not change with the increase in test order, but for scenarios with developing mental schema (L1 and L2 scenario), the small-worldness metric would increase with training times, indicating the strong interaction between the completeness of mental schema development and mental workload.

## 4. Discussion

It is believed that oscillations in ongoing brain activity underlay periodicity in sensorimotor processing and can represent temporal predictions via their phase dynamics [42]. In this study, we investigated the influence of mental schema evolution and task difficulty on the mental workload of UAV operators from the perspective of phase synchronization.

When the operator manually controls the micro-UAV, he should perform target-oriented navigation within dynamic physical limits and error margins [43], in which sensorimotor predictions are essential for adaptive behavior. Neuronal oscillations imply a mechanism for sampling sensory information and motor actions rhythmically, and the phase dynamics appear to be highly adaptable to temporal predictability in the environment while actively using the time processed by neurons to optimize task performance [42]. Because none of the participants had any experience in operating UAVs on Test1, they were unable to maneuver the aircraft as they wished. At this point, they tried to explore the possible brain regions involved in the task, resulting in significantly higher phase synchrony than the other three tests (Test2–Test4). Additionally, after repeated training and testing tasks, they became more efficient at deriving from the disuse of extraneous brain regions to obtain better performance [8], resulting in a decrease in phase synchrony among brain regions. This is further supported by the finding that the strength of the connection on the first day increased with the difficulty of the task, as it required more effort to complete the task.

During the operation of the UAV, the operator needed to effectively complete the whole process of perception, cognition, and action execution. Global and local efficiency measure how efficiently information is exchanged at the global and local levels, respectively [25]. Integrated (or distributed) information processing would benefit from efficient information transfer throughout the network across the brain [14]. As shown in Figure 4A, because the training tasks (on day 2 and day 3) were performed only in the L0 scenario, they gradually developed the mental schema for that scenario, while in other scenarios (L1 and L2), the addition of wind force would make established mental schema become inadequate, resulting in a decline in global efficiency. Moreover, when there was no mental schema (Test1) or mental schema was developing (Test2), they could not consciously form precise strategies to complete tasks in each scenario, so there was no consistent trend of global efficiency with the increase in task difficulty. However, after the formation of a complete mental schema (Test3 and Test4), the global efficiency decreased with the increase in task difficulty, which may be the result of the deviation in the mental schema caused by the increase in difficulty. These observations demonstrate the preliminary relationship between mental schema, global efficiency, and mental workload.

In order to capture dynamic changes underlying the reconfiguration of the brain network, the local efficiency metric was further analyzed. As shown in Figure 4B, nodes with high local efficiency were concentrated in the frontal region in Test1, and with repeated training and testing, the gravity center of local efficiency distribution moved to the posterior region. Previous studies have demonstrated that when brain networks remained small-world modular, functional connectivity patterns could be flexibly adjusted to meet mental demands [44,45,46]. In situations where highly adaptive control was required, the functional connectivity patterns of the frontal–parietal networks varied more than other networks, and these connection patterns could be used to identify the current task demands [44,47,48]. In addition, frontal cortex activity, which underpins skill learning, follows an anterior-to-posterior direction based on the association–motor hierarchy of motor preparation and control [49,50], which corresponds to the transition from the level of executive control to the level of motor control, thereby reducing the need for central resources [51,52]. These results provide additional evidence for brain changes associated with neural adaptation prior to motor skill automation.

The formation of the small-world structure is the result of natural selection to meet the balance between low consumption of neuronal resources and efficient information communication, which is a process by which brain networks evolve to adapt to the demands of the task [8,14,48]. From the perspective of network dynamics, small-world structural networks support rich dynamical behaviors and adaptive functions, such as the rapid propagation and integration of information [49], flexible transitions of functional connectivity patterns [50], and critical-like state for fast response to external demands [42,51,52], all of which are characteristics of skill level development. Therefore, as the skill level increased, the mental schema of operating the UAV was gradually developed, and the small-world characteristics of functional connectivity networks became stronger, but the change in difficulty levels did not cause the change in small-world characteristics. Furthermore, as shown in Figure 4C, the small-worldness metric showed a consistent increase from Test1 to Test4 for L1 and L2, but not for L0, indicating that the brain was still learning to adapt to the more difficult scenarios, even after forming a stable spatial pattern of local connections.

Our results are in line with some research evidence in which large-scale network reconfigurations in relation to motor learning have been observed in task-based studies [53,54]. The increased efficiency of communication between different specialized regions was found in a submarine navigation training task by Fallani et al. [55], like the mission in this study, which involved both cognitive and visuomotor engagement. Such a small-world topology could be a need for new strategy exploration and formation [55,56], and differences in strategy are reflected in the connection pattern in the theta band [57]. Although the impact of increased task difficulty on network efficiency has not been consistently concluded as mentioned earlier, our results still support our previous findings in which indicators of mental workload could be the result of mental schema development biased by task difficulty [58]. In summary, we found that the global and local efficiency of the network decreased with the increase in task difficulty after the formation of developed mental schema. Furthermore, the development of mental schema may lead to the dynamic reconfiguration of the functional connectivity network of the brain, including the shift in the distribution center of local efficiency to the posterior areas of the brain and the increase in the small-world attribute, making the network more flexible to meet highly adaptive control.

### Limitations and Future Research

There are still several limitations to be addressed. Firstly, only the phase synchronization in the theta frequency band was investigated, and it has been suggested that oscillations in different frequency bands have different functions and their phase dynamics are critical for sensation, perception, and visuomotor engagement [42]. Secondly, the universality of the findings from this study in other motor training tasks should be tested. Thirdly, due to the large individual differences in UAV operation capabilities, the individual differences in brain topology need to be further explored. We are working on solving these problems in a future study. Finally, only male participants were recruited in this study, leading to a significant gender imbalance in data collection. We will further verify whether our findings also apply to female subjects in the future.

## 5. Conclusions

In conclusion, we demonstrated the unique contributions of the brain network analyses of neuroimaging data to explore the effect of mental schema development and task difficulty on the mental workload, especially for skill-based tasks such as manually flying a quadrotor UAV. It was found that during different tests, the global efficiency did not change, but the gravity center of the local efficiency of the network moved from the frontal to the posterior area; the small-worldness of the functional brain network became stronger. After forming the developed mental schema, the global and local efficiency of the network decreased with the increase in task difficulty, showing sensitivity to the increase in mental workload. Our results demonstrate brain network reconfiguration during the learning process of a task and reveal the importance of the developed mental schema for the assessment of mental workload. We argue that a stable spatial pattern of local efficiency announced the formation of mental schema, and the global and local efficiency of the brain network are associated with mental workload with developed mental schema.

## Figures and Tables

**Figure 1 brainsci-13-00373-f001:**
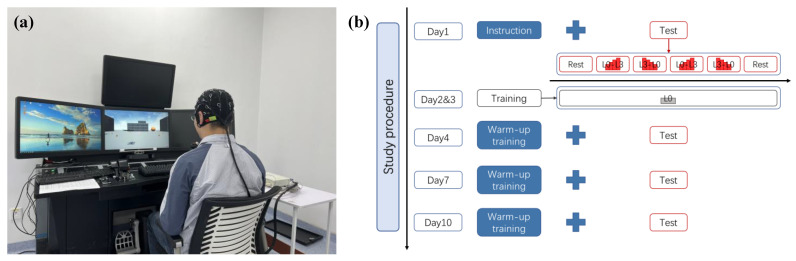
Illustration of the experimental environment, task scenarios, and experimental procedure arrangements. (**a**) Simulated UAV operation tasks and experimental environment. (**b**) Schematic of the study procedure (adopted from Gu et al., 2022 [32]).

**Figure 2 brainsci-13-00373-f002:**
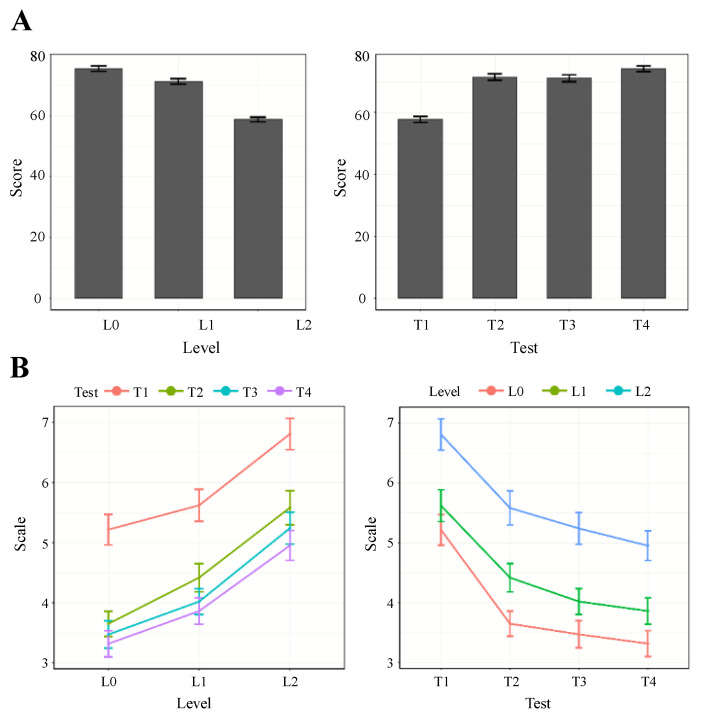
Summary of performance and subjective ratings across different conditions. (**A**) Behavioral results of UAV task. (**B**) Perceived workload result of UAV task. The error bars indicate the standard error of the mean (SEM).

**Figure 3 brainsci-13-00373-f003:**
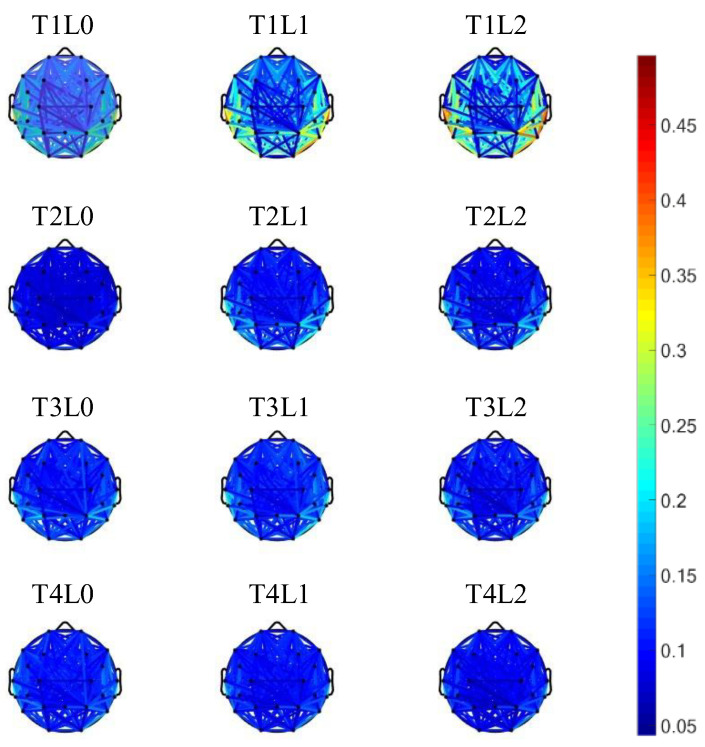
Group average connection strength of channel pairs with significant interaction across different conditions (T for Test, L for Level).

**Figure 4 brainsci-13-00373-f004:**
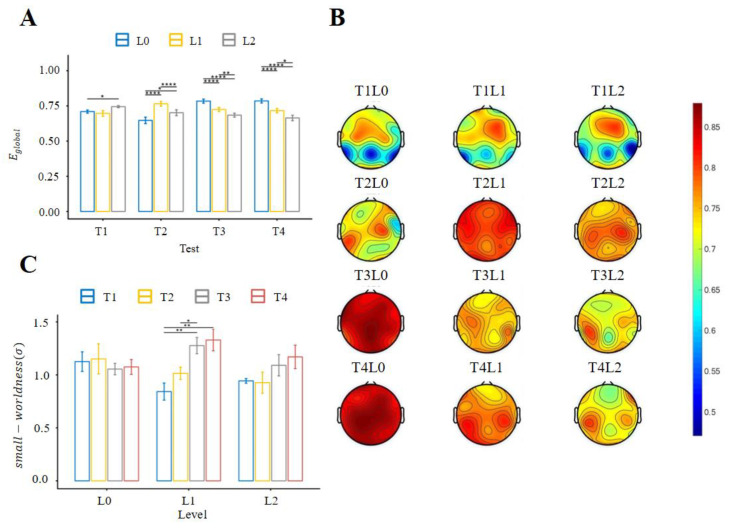
Statistical results of brain network metrics as the task difficulty and test order changed. (**A**) Global efficiency. (**B**) Grand average local efficiency for each channel (T for Test, L for Level). (**C**) Small-worldness. (*: *p* < 0.05, **: *p* < 0.01, ****: *p* < 0.0001 after correction). The error bars indicate the standard error of the mean (SEM).

## Data Availability

The data presented in this study are available on request from the corresponding author. The data are not publicly available due to restrictions of privacy.

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
