# Peer review of "Efficiency of the Brain Network Is Associated with the Mental Workload with Developed Mental Schema"

_brainsci, 2023, doi:10.3390/brainsci13030373_

Round 1

Reviewer 1 Report

Overall I think it is a great written paper.I would love to see the authros to address the following issues before publication :

1) From line 96-line97, since the global efficiency , local efficiency and small-worldness are important metrics measured in this experiment, it will be great if the authors can also add one or two sentences to provide clear definitions (narratively/ conceptually) of those three metrics in the introduction.

2) Fig1 b) is not intuitive. The first impression of Fig1b) is something like the training has 4 different difficulty level. Re-organize the figure is recommended so future readers will not get confused. 

3)Line 171, can authors explain why choosing 3 scenarios rather than 4 scenarios here? From the experiment design, it seems that there are 4 scenarios. At this step , I believe we do not need to consider the real number of scenarios the authors used in the analysis. 

4)Line 165-line 189, it will be much more straightforward for readers if the notation can be explained the first time it shows up in the formula. For example. Min{L_ij} firstly shows up in formula 2), but gets explained after formula (5) .

5)  In line 202, since L3 scenario is part of the design, it will be great if the authors can provide more details about the assessment results, such as mean/range/variance etc, or maybe a bar plot with error bars. In that case, future researchers can make their own decision if they want to replicate the whole experiment design(with four conditions) or just use three of them. 

6)In line 218-220, it looks like some of the df are sphericity assumed and some of the df has been Greenhouse-geissor corrected. Just want to double check with the authors that all df reported here in the paper are G-G corrected. 

7) Line 233, main effect rather than “man effect”

8) For Fig 4. A and C, is there a particular reason the authors switched the scale? In Fig4.A, the X-axis is T1-T4 and the categories are L0-L2, in Fig4.B, it’s opposite 

9) In Discussion part, the authors mentioned that “the mental schema of operating the UAV was gradually developed .. “. The word mental schema appears multiple times in the article. I wonder if the author mentioned the definition of mental schema somewhere in the paper, and if there is any scale/scores the authors used to indicate “the mental schema development”. If there is, in which case (any quantified metric) it is considered as developed.   

10) In line 393-395, the authors mentioned that “the global efficiency did not change, but the gravity center of the local ….” . Any statistics evidence showed that(“the global efficiency did not change”) in the results?  It will be great that the authors clearly mentioned that in the results before adding them into the conclusion.

Reviewer 2 Report

A broader literature review of these studies should be done. At the same time, it proposes to correlate the results with other authors and research methods. Please also describe the research groups in more detail.

Reviewer 3 Report

The authors conducted a brain network analysis using electroencephalography (EEG) to evaluate the relationship between brain network change and mental workload along with mental schema development.

The results obtained in this study are interesting, valuable, and worthy of publication. However, several points need to be addressed before publication can be considered.

1.     The authors showed that the global and local efficiency of the brain network decreases with increasing task difficulty during mental schema development. Based on the results, they concluded that this pattern of change in the brain network might be an indicator of mental workload. However, this result only led to the conclusion that task difficulty is related to changes in the brain network associated with mental workload. To conclude that the efficiency of the brain network is an indicator of mental workload, it is necessary to show the correlation between task-related parameters and brain network efficiency. In light of the foregoing, it is necessary to conduct further analyses or withdraw the conclusion about brain network efficiency as an indicator of mental workload.

2.  While the Introduction section is overly verbose, the purpose of this study is not clearly stated. The contents presented in lines 102 to 109 are not the purpose of the research, but rather, the summary of the research conducted by the authors. The section should be modified to express the purpose of the research clearly.

3.  Is there a reason why all the participants in this study are males? It is crucial for the authors to indicate their reasons for excluding females and to consider whether those reasons are justifiable.

4.  What is the premise for designing the task difficulty level in an ascending and descending order rather than a random order in the test session? If this method is intended to measure the gradual change of the network related to the development of mental schema (such as measuring the input-output curve), it seems necessary to discuss this point for better understanding.

5.  The authors excluded the data of 27 participants in the analysis. The 27 participants scored 0 in Test 1 due to insufficient operational skills. What does “insufficient operational skills” mean? Had “insufficient operational skills” been planned as an exclusion criterion from the analysis at the planning stage of this study? In addition, the authors need to explain why the exclusion of 27 participants in the analysis is justifiable.

6.  L3 level data were also excluded from the analysis. The reason those data were excluded from the analysis seems difficult to justify.

7.  Since the mental schema is formed through motor learning, it is recommended that the authors add the relevance of the study results to motor learning in the discussion.
